# Single-cell transcriptomics reveals immunosuppressive microenvironment and highlights tumor-promoting macrophage cells in Glioblastoma

**Han Cheng‡, Yan Yan, Biao Zhang, Zhuolin Ma, Siwen Fu, Zhi Ji, Ziyi Zou, Qin Wang◯ \***

Department of Clinical Laboratory, Tianjin Key Laboratory of Cerebral Vascular and Neurodegenerative Diseases, Tianjin Neurosurgical Institute, Tianjin Huanhu Hospital, Tianjin, China

‡ The First Author.
\* tjhhhospitallab@126.com

## Abstract

Glioblastoma (GBM) is the most prevalent and aggressive primary brain malignancy in adults. Nevertheless, the cellular heterogeneity and complexity within the GBM microenvironment (TME) are still not fully understood, posing a significant obstacle in the advancement of more efficient immunotherapies for GBM. In this study, we conducted an integrated analysis of 48 tumor fragments from 24 GBM patients at the single-cell level, uncovering substantial molecular diversity within immune infiltrates. We characterized molecular signatures for five distinct tumor-associated macrophages (TAMs) subtypes. Notably, the TAM_MRC1 subtype displayed a pronounced M2 polarization signature. Additionally, we identified a subtype of natural killer (NK) cells, designated CD56$^{dim}$_DNAJB1. This subtype is characterized by an exhausted phenotype, evidenced by an elevated stress signature and enrichment in the PD-L1/PD-1 checkpoint pathway. Our findings also highlight significant cell-cell interactions among malignant glioma cells, TAM, and NK cells within the TME. Overall, this research sheds light on the functional heterogeneity of glioma and immune cells in the TME, providing potential targets for therapeutic intervention in this immunologically cold cancer.

## Introduction

GBM is a malignant brain tumor characterized by its heterogeneous cell populations, which hinder the efficacy of standard treatments and immunotherapies [1]. The current standard treatment regimen includes maximal surgical resection, followed by radiotherapy and chemotherapy. Despite these interventions, the prognosis for patients remains poor, with a median survival time of approximately 15 months and a 5-year survival rate below 5% [2]. Immunotherapy has a certain degree of targeted selectivity for cancer cells, killing cancer cells while preserving normal brain tissue, making it a potential treatment strategy for GBM. The CheckMate-143 trial was the first phase III study to propose the use of the PD-1 inhibitor nivolumab for treating GBM patients. Still, the median overall survival (OS) was comparable between the nivolumab and bevacizumab treatment groups, showing no

**Data availability statement:** All relevant data are within the manuscript and its Supporting Information files.

**Funding:** The author(s) received no specific funding for this work.

**Competing interests:** The authors have declared that no competing interests exist.

clinical application value [3]. The CheckMate-498 trial evaluated the efficacy of nivolumab combined with radiotherapy versus temozolomide (TMZ) combined with radiotherapy in newly diagnosed GBM patients with unmethylated O6-methylguanine-DNA methyltransferase (MGMT) status after surgical resection of the tumor [4]. In a Phase I clinical trial called INCIPIENT, the CARv3-TEAM-E T cell therapy administered via intraventricular injection successfully led to significant tumor reduction in three patients with recurrent GBM, indicating the promising therapeutic potential of this novel CAR-T cell treatment in patients with recurrent GBM [5]. Although immunotherapy has brought new hope and treatment strategies to GBM therapy, it demonstrates resistance to various immunotherapies, including anti-PD1/PD-L1 therapies, chimeric antigen receptor CAR-T, CAR-NK, and macrophage-based treatments. This resistance is primarily due to the highly immunosuppressive TME and the presence of immune checkpoint molecules that inhibit effective immune responses [1,2,6]. Consequently, there is an urgent need to elucidate the mechanisms underlying the immunosuppressive TME in GBM, which could lead to more effective therapeutic strategies.

Various immune cell types and multiple functional states are important parameters associated with responsiveness to immunotherapies [7]. TAM cells are pivotal in regulating tumor progression and immune evasion, making them promising therapeutic targets [8,9]. However, their functional and molecular heterogeneity poses significant challenges due to the high plasticity of TAMs within the TME. In GBM, macrophage populations include embryonic yolk sac-derived brain-resident macrophages (microglia) and bone marrow-derived macrophages (BMDMs) that are recruited after tumor initiation [10,11]. Recent studies suggested that the traditional M1/M2 cell states observed in vitro may not accurately represent the TAM cell states in vivo [12]. Glioma-associated myeloid cells and glioma cells release cytokines and metabolites that inhibit the function of tumor-infiltrating lymphocytes (TIL) [13,14]. The limited presence of TILs (less than 5%) alongside immunosuppressive myeloid cells in GBM categorizes it as an 'immune cold' tumor, hindering the effectiveness of immunotherapies [15,16]. Therefore, shifting myeloid cells towards an anti-tumorigenic phenotype could enhance T cell infiltration and activation, promoting anti-tumor immunity in GBM. A comprehensive understanding of the cellular and molecular dynamics within the GBM is essential to develop such strategies. Without these insights, the current immunotherapies will likely face challenges in effectively treating GBM patients.

NK cells, essential components of the innate immune system, play a critical role in suppressing tumors by recognizing and eliminating cancer cells without prior sensitization [17]. These cells directly target and eliminate cancer cells through receptors that detect stress markers on tumor cells, facilitating a rapid immune response [18,19]. Moreover, NK cells enhance adaptive immunity by releasing cytokines and chemokines that recruit and activate other immune cells [20]. However, the efficacy of NK cells is often hindered by the immunosuppressive TME, which can impair NK cell function and survival. In tumor tissue, NK cells frequently exhibit an exhausted state, characterized by decreased cytotoxic activity and diminished cytokine production [21]. NK cells have a complex array of activating and inhibitory receptors. Within the TME, the balance tends to lean towards inhibition due to an abundance of ligands for inhibitory receptors on tumor cells or a lack of sufficient activating signals [22]. The TME contains various suppressive elements, including regulatory T cells (Tregs), myeloid-derived suppressor cells (MDSCs), and immunosuppressive cytokines like TGF-β and IL-10. These components collectively hinder NK cell functions and contribute to their exhaustion [2,23]. A deeper understanding of the TME could shed light on the status of NK cells, offering a theoretical foundation for enhancing their functionality and improving outcomes in immunotherapy.

The rapidly evolving single-cell RNA sequencing (scRNA-seq) has significantly advanced the identification of cellular populations and functional states within the TME [24]. So, scRNA-seq offers unprecedented opportunities to elucidate the cell components and functional states in GBM TME. In our study, we analyzed 48 clinical samples from 24 GBM patients using scRNA-seq to investigate the cellular composition and communication networks among cell types. Our results revealed significant heterogeneity among malignant glioma cells and immune cell types, such as TAM, T cells, and NK cells. We identified molecular signatures for five distinct TAM subtypes, with the TAM_MRC1 subtype showing an M2-like polarization signature associated with a tumor-promoting role. Furthermore, we found a subset of NK cells exhibiting an exhausted phenotype characterized by reduced cytotoxic activity and elevated stress signatures. These findings highlight the intricate interactions within the GBM TME and offer insights into potential therapeutic strategies for the future.

## Methods

### Ethical statement

In our study, the scRNA-seq datasets were collected from two published studies with open-assess, so the ethical declaration is not applicable. However, ethical approvals for the original studies were obtained, and details can be found in the respective publications. For the datasets from Abdelfattah et al. [2], human tumor tissues were obtained under Institutional Review Board (IRB)-approved protocols (Pro00014547) at Houston Methodist Hospital, Houston, Texas and MD Anderson Cancer Center (PA 19-0661) following national guidelines. All patients signed informed consent during clinical visits before surgery and sample collection. For the datasets from Ravi et al. [25], the local ethics committee of the University of Freiburg approved the data evaluation, imaging procedures, and experimental design (protocol 100020/09 and 472/15_160880). These studies were approved by an institutional review board.

### Data collection

Single-cell transcriptome profiles of GBM were collected from two publicly available datasets. The scRNA-seq dataset from the study by Abdelfattah et al. [2] contains 44 tumor fragments from 18 glioma patients, including two low-grade gliomas (grades II), 11 ndGBM (grade IV) and 5 rGBM (grade IV). To minimize the patients' variance, we downloaded the raw fastq sequences from 40 high-grade GBMs datasets from GSE182109. We also collected a scRNA-seq dataset from Ravi et al. [25], where lymphoid and myeloid populations (CD45+/CD3+) were sorted from eight patients diagnosed with de novo glioblastoma. We downloaded the matrix Seurat object from https://osf.io/4q32e/https://doi.org/10.17605/OSF.IO/4Q32E.

### Single-cell RNA sequencing analysis

The raw Illumina sequencing reads from 40 samples sequenced by Abdelfattah *et al.* were aligned to GRCh38 (human) using Cell Ranger (V7.2.0) software with default parameters. Subsequently, genes were quantified as UMI counts using Cell Ranger. Downstream analysis was performed on filtered feature counts generated by Cell Ranger, and low-quality single cells containing < 200 expressed genes or > 20% mitochondrial transcripts were removed. Additionally, genes expressed in fewer than three single cells were removed.

### SoupX and DoubletFinder remove potential droplets

SoupX (V1.6.0) [26] was used to estimate and remove cell-free mRNA contamination in droplet-based sc-RNAseq data. The raw_feature_bc_matrix and filtered_feature_bc_matrix

resulted results from Cell Ranger were used. Additionally, we identified potential single-cell doublets using DoubletFinder (V2.0.3) (https://github.com/chris-mcginnis-ucsf/Doublet-Finder), with an expectation of a 7.5% doublet rate assuming Poisson Statistics.

## Clustering and cell type identification

Then scRNA-seq data from different samples were merged into Seurat (V4.2.0) [27] R package, then normalized using the *NormalizeData* function and scaled on 2000 most variable features using the *FindVariableFeatures* and *ScaleData* functions. Computed principal components were batch corrected for sample variations using the Harmony R package (V1.0). We used batch-corrected PCs as input for Louvainbased graphing and chose resolution parameters with 0.5, 0.8 and 1.0. *FindAllMarkers* function was used to identify cluster-specific marker genes and visualization with dot and feature plots. The genes expressed in each cluster and SingleR (V1.4.1) [28] were examined to identify the cell types.

## GO and KEGG enrichment

We used clusterProfiler [29] package to identify and visualize enriched pathways in our subsets. Differentially expressed genes (DE genes) between cell types were identified with FindMarkers or FindAllMarkers function with default parameters. Subsequently, we employed the enrichGO and enrichKEGG functions to identify enriched pathways for Gene Ontology (GO) enrichment analysis and the Kyoto Encyclopedia of Genes and Genomes (KEGG) analysis, respectively.

## CNV estimation

Copy-number variations (CNVs) were estimated by the inferCNV software package (https://github.com/broadinstitute/inferCNV). The software aligns genes to their chromosomal location and applies a moving average to the relative expression values, with a sliding window of 100 genes within each chromosome. Malignant glioma cells from some patients were used as observation datasets, and T_NK and myeloid cells (randomly selected 2000 cells for each cell type) were used as normal reference set.

## Definition of cytotoxicity, inflammatory, and stress gene sets

To evaluate the functional differences among NK cell subsets, we created an extensive compilation of cytotoxicity, inflammatory, and stress-related gene sets and activating and inhibitory receptors from previous research. The gene sets mentioned in Zhang et al.'s [30] article (refer to S2 Table for more information) were included. We computed the scores for each gene set using the *AddModuleScore* function in Seurat.

## Receptor ligand interaction analysis

To investigate potential interactions among various cell types within the GBM TME, a cell-cell communication analysis was conducted using CellPhoneDB (V3.0) [31]. This analysis included cells annotated as glioma, five TAM subtypes and four NK subtypes. CellPhoneDB facilitated the assessment of cell-cell interactions by analyzing enriched receptor-ligand interactions, which depend on the expression of a specific receptor by one cell type and its corresponding ligand by another. Subsequently, we selected interactions that were biologically significant.

## Single-cell regulatory network analysis

To identify key transcription factors (TFs) across various cell types, a cis-regulatory analysis was conducted using pySCENIC (V0.11.2) [32,33]. The analysis focused on five subtypes of

TAM. TFs were identified utilizing GENIE3, grouped into modules known as regulons, and analyzed via RcisTarget employing default gene-motif rankings. Subsequently, regulon activity for each cell in the dataset was assessed using AUCell.

## Results

### Dataset description

Single-cell transcriptome profiles of GBM were collected from two publicly available datasets. The scRNA-seq dataset by Abdelfattah et al. consisted of 40 tumor fragments from 16 GBM patients, including 11 newly diagnosed (termed as ndGBM in our research) and 5 recurrent (termed as rGBM in our study) cases. Additionally, we incorporated a scRNA-seq dataset from Ravi et al., which involved sorting lymphoid and myeloid cell populations (CD45+/CD3+) from eight patients with *de novo* GBM. Clinical details of the GBM patients are provided in S1 Table. The clinical information was sourced from an open-access database and did not require an ethical declaration. Ethical approval was obtained for the original studies, with details available in the respective publications. To ensure data quality, cells with low read depths or high mitochondrial gene expression were excluded, and potential doublets were removed using the SoupX and DoubletFinder R packages as detailed in 'Method' section. In total, our analysis included 252,452 high-quality cells, with 211,376 cells from the Abdelfattah et al. dataset and 41,076 cells from the Ravi et al. dataset.

### Main cell type annotation

After the removal of the batch effect between samples, 252,452 single cells were clustered into 33 clusters with a resolution value of 0.8 in the *FindClusters* function. Cell types were annotated based on classic markers identified in previous studies and the SingleR package. These cell types included B cells, characterized by the expression of *CD19* and *CD79A*. T_NK cells, comprising both T and NK cells, identified by markers such as *CD3D/G* and *NCAM1*. Myeloid cells, expressing *LYZ*, *ITGAM*, and *CD14*. Stromal cells including pericytes (expressing *ACTA2* and *PDGFRB*), endothelial cells (expressing *PECAM*), and oligodendrocytes (expressing *OLIG2* and *MBP*). Additionally, malignant glioma cells were characterized by the expression of *SOX2*, *OLIG1*, *GFAP*, and *S100B* (Fig 1A–1C). Cells from two datasets were integrated well (Fig 1D). We quantified the relative tissue enrichment of major cell types in each sample and dataset (Fig 1E). Consistent with prior studies, the proportions of cancer, stromal, and immune cells varied significantly across samples, potentially due to inherent differences in tumor phenotypes or the specific locations within the tumor from which biopsies were taken. Specifically, in the specimens from Ravi et al., nearly all identified cells were of immune origin, with only a minimal presence of stromal cells, except in the sample NC_P234_JAK (Fig 1E, S1A Fig). All samples demonstrated a pronounced enrichment of myeloid cells, whereas the fractions of T_NK and B cells were relatively low. In contrast, the samples from the study by Ravi et al. predominantly contained glioma and myeloid cells, with other immune cells comprising approximately 10.6% of all cells analyzed. Notably, three fragments from ndGBM_01 (ndGBM_01_A, ndGBM_01_C, and ndGBM_01_D) and one from ndGBM_06 exhibited more than 80% glioma cells. Additionally, samples from recurrent GBM cases (rGBM_03, rGBM_04, and rGBM_05) contained higher proportions of glioma cells compared to their newly diagnosed counterparts (Fig 1E). These results align with previous findings from flow cytometry and single-cell analyses [2,11,12], validating the accuracy of our cell type identification.

### Inter- and intra-tumoral molecular heterogeneity of glioma cells

To explore the molecular heterogeneity of malignant glioma cells, we conducted *de novo* clustering of these cells into 34 distinct clusters (S1B Fig). Consistent with expectations, glioma

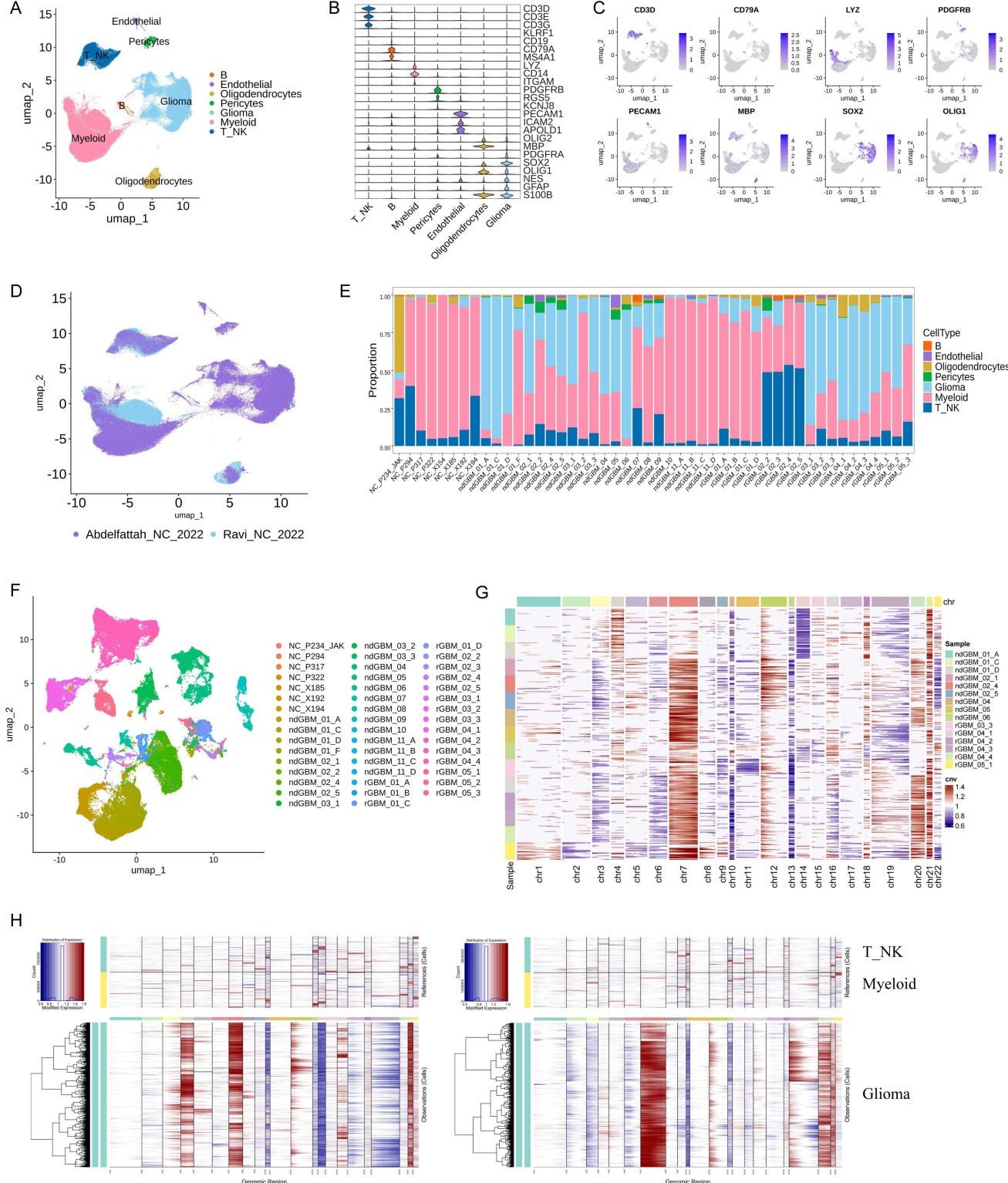

**Fig 1. Dissection of the tumor microenvironment in human GBM with scRNA-seq.** (A) UMAP plots of cells from the 48 tumor fragments profiled in this study, with each cell color coded to indicate the associated cell types. (B) The violin plot displayed the classic markers used for cell type annotation. (C) FeaturePlot displayed the classic markers used for cell type annotation. (D) UMAP plot of cells from the 48 tumor fragments profiled in this study, with each cell color coded to dataset. (E) Major cell type composition of each tumor fragment and dataset. (F) UMAP plot of malignant glioma cells, with each cell color coded to fragments. (G) Heatmap of CNA profiles inferred from scRNA-seq of tumor cells. Red indicated genomic amplifications,

and blue indicated genomic deletions. (H) Heatmap of CNA profiles inferred from scRNA-seq of tumor from ndGBM_04 (left) and rGBM_O4_1 (right). T_NK and Myeloid cells were used as reference.

cells exhibited patient-to-patient heterogeneity, similar to many other cancer types (Fig 1F). We carried out a CNV analysis using inferCNV on 15 GBM fragments, each containing 2000 glioma cells. T_NK and Myeloid cells (randomly selected 2000 cells for each cell type) were used for reference. This analysis revealed shared and fragment-specific mutations across different regions within the same patient and among different patients (Fig 1G), aligning with previously documented inter- and intra-tumoral genomic heterogeneity in GBM [2,25]. Notably, all fragments showed common alterations such as loss of chromosomes 10/10q, 13/13q, and gain of chromosome 7/7q, which are recurrent copy number changes in human GBMs [34]. While major copy number changes were typically consistent across different fragments within each tumor, unique indels, and mutational patterns also distinguished individual fragments in each patient (Fig 1H). These findings corroborate the exome sequencing results of Abdelfattah et al. [2].

## Functional heterogeneity of TAM in GBM

Myeloid cells, including microglia and BMDM, constitute the predominant stromal compartment in the TME of GBM. To elucidate the cellular and molecular diversity within myeloid populations, we analyzed 114,351 myeloid cells through *de novo* clustering. This approach enabled the identification of three distinct myeloid cell subtypes characterized by unique gene expression profiles (Fig 2A). TAM was noted for expressing well-known microglia markers such as *P2RY12*, *TMEM119*, *BHLHE41*, *SORL1*, *SPRY1*, and *SRGAP28*, alongside classical macrophage markers including *C1AC*, *C1QA*, and *APOE*. Within the BMDM subset, the cDC cell type is marked by the expression of traditional dendritic cell markers, including *CD1C*, *CLEC9A*, *CLEC10A*, and *LAMP3*. Additionally, monocyte-derived cells were distinguished by high levels of *S100A8*, *S100A9*, and *VCAN* (Fig 2B). All patients and fragments contributed to each myeloid cell type.

TAMs were further subjected to a second round of *de novo* clustering. In contrast to the traditional *in vitro* classifications of M1- or M2-like macrophages, TAMs in GBM displayed diverse behaviors *in vivo* (S2A–S2C Fig), consistent with observations in other cancer types [12,35]. To identify distinct TAM subtypes, we annotated these subtypes based on lineage markers (Fig 2C, 2D). The TAM_MRC1 subtype was defined by high levels of *MRC1*, an established M2 polarization marker, and other M2-associated markers like *CD163* and *CD204*/MSR1. Previous research has highlighted the role of *MRC1*, also known as ITGAM, in driving M2 polarization of macrophages potentially through the FAK/Akt1/β-catenin signaling pathway [36]. On the other hand, the TAM_ISG15 subset exhibited increased levels of *ISG15*, along with markers linked to M1 polarization, such as *CD86*, *CXCL9*, *CXCL10*, *IRF1*, and *CD40*. Interestingly, almost all identified clusters and subtypes co-expressed *SPP1*, *C1QA*, *C1QB*, and *C1QC* genes (Fig 2D). This discovery contrasts with findings in colorectal cancer and breast cancer, where *SPP1* expression typically shows mutual exclusivity with *C1QC* expression [8].

The functional phenotypes of TAMs are generally classified within an in vitro M1/M2 dual polarization framework. Subsequently, the *AddModuleScore* function from the Seurat package was employed to quantify the signatures of M1, M2, and pro-inflammatory gene sets (S2 Table). Notably, TAM_ISG15 macrophages displayed relatively higher M1 signatures, whereas TAM_MRC2 macrophages exhibited relatively higher levels of canonical M2 signatures, as expected. Additionally, TAM_NLRP3 macrophages demonstrated enhanced

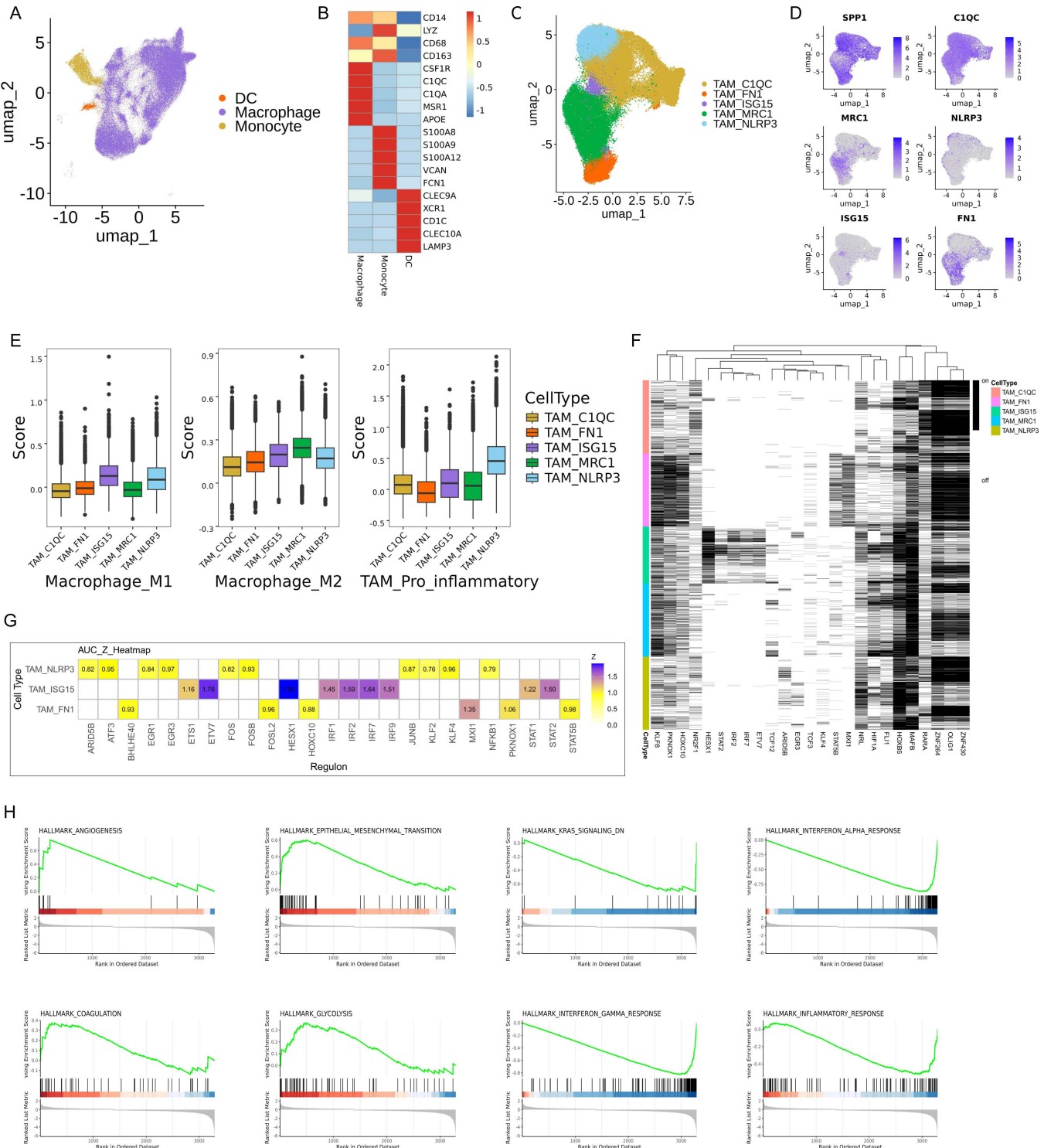

**Fig 2. Cellular composition of myeloid cells.** (A) UMAP plots of myeloid cells, with each cell color coded to indicate the associated cell types. (B) Heatmap displayed the classic markers used for myeloid cell type annotation. (C) UMAP plots of five TAM subtypes, with each cell color coded to TAM subsets. (D) FeaturePlot displayed the classic markers used for TAM subtype annotation. (E) M1-like, M2-like macrophage, and pro_inflammatory scores of five TAM cell subtyps. (F, G) Key transcription factors across five TAM subsets and a cis-regulatory analysis using pySCENIC. (H) Different enrichment of 'Hallmark' pathways between TAM_MRC1 and TAM_ISG15 by GSEA analysis.

pro-inflammatory signatures (Fig 2E). These findings challenge the adequacy of the conventional in vitro polarization model and underscore the complexity of TAM phenotypes within the TME.

We subsequently analyzed the regulatory networks underlying each TAM subset using the SCENIC. We identified specific TF regulons for each TAM subset (Fig 2F, 2G). For instance, the TAM_ISG15 subset exhibited upregulation of positive regulators, including ETV7, HESX1, IRF1, IRF2, IRF7, and IRF9, suggesting its potential role in enhancing immune responses. Similarly, the TAM_NLRP3 subset showed increased activity of ATF3, ARID5B, ECR1, and ECR3. Notably, in the TAM_MCR1 subset, the KLR8 and NRL regulons were significantly activated. These findings suggest potential candidates that could drive the functional differences observed among TAM subsets, contributing to a deeper understanding of their heterogeneity and functions.

To investigate the association of TAM_MRC1 and TAM_ISG15 subtypes with specific signaling pathways, we performed GSEA analysis using the Hallmarks gene set from the MsigDB. Our analysis indicated that the TAM_MRC1 subtype was significantly enriched in pathways such as HALLMARK_ANGIOGENESIS, HALLMARK_EPITHELIAL_MESENCHYMAL_TRAN-SITION, HALLMARK_COAGULATION, and HALLMARK_GLYCOLYSIS. On the other hand, the TAM_ISG15 subtype exhibited significant enrichment in HALLMARK_INTER-FERON_ALPHA_RESPONSE, HALLMARK_INTERFERON_GAMMA_RESPONSE, HALL-MARK_KRAS_SIGNALING_DN, and HALLMARK_INFLAMMATORY_RESPONSE (Fig 2H). It is well known that M2-like macrophages are recognized for secreting various factors that stimulate angiogenesis, including Vascular Endothelial Growth Factor (VEGF), Platelet-Derived Growth Factor (PDGF), and Transforming Growth Factor-Beta (TGF-β). These factors enhance endothelial cell proliferation and migration, crucial processes in angiogenesis [37]. Epithelial-mesenchymal transition (EMT) is a key biological mechanism where epithelial cells transition into a mesenchymal state, involving loss of cell-cell adhesion and acquisition of migratory and invasive characteristics. This transformation is vital in cancer progression as it aids invasion and metastasis. M2 macrophages play a role in promoting EMT by releasing factors like TGF-β and IL-10, which induce EMT in nearby epithelial cells. This reciprocal interaction establishes a feedback loop where EMT supports M2 macrophage polarization, and conversely, M2 macrophages enhance EMT, contributing to a more aggressive tumor phenotype [38,39].

## Detailed reclustering of T and NK cells

T and NK cells play critical roles in regulating tumorigenesis and cancer progression, but their characterization in GBM remains elusive. In our study, we classified the T and NK cells into five distinct subtypes based on specific cellular signature markers (Fig 3A). These subtypes include T cells (n = 26,193), characterized by the expression of *CD3D* and *CD3G*. Two NK subtypes, NK_CD56$^{dim}$ (n = 1,345) and NK_CD56$^{bright}$ (n = 993), identified by the relative expression levels of *NCAM1* (CD56) and *FCGR3A* (CD16). NKT cells (n = 613) exhibited markers of both T and NK cells. ILC cells (n = 97), distinguished by the expression of *KIT* (Fig 3B). Consistent with previous studies, we found significant variability in the proportions of these cell types across samples, with T cells being a substantially larger fraction than the other four cell types (Fig 3C).

## Detailed reclustering of CD4 + T and CD8 + T cells

To analyze T cells, we performed two rounds of unsupervised clustering. Initially, we distinguished between CD4 + T (CD4T) and CD8 + T (CD8T) cells based on the expression of *CD4* and *CD8A* (Fig 3D). Subsequently, the CD4 + T cells were further categorized into three

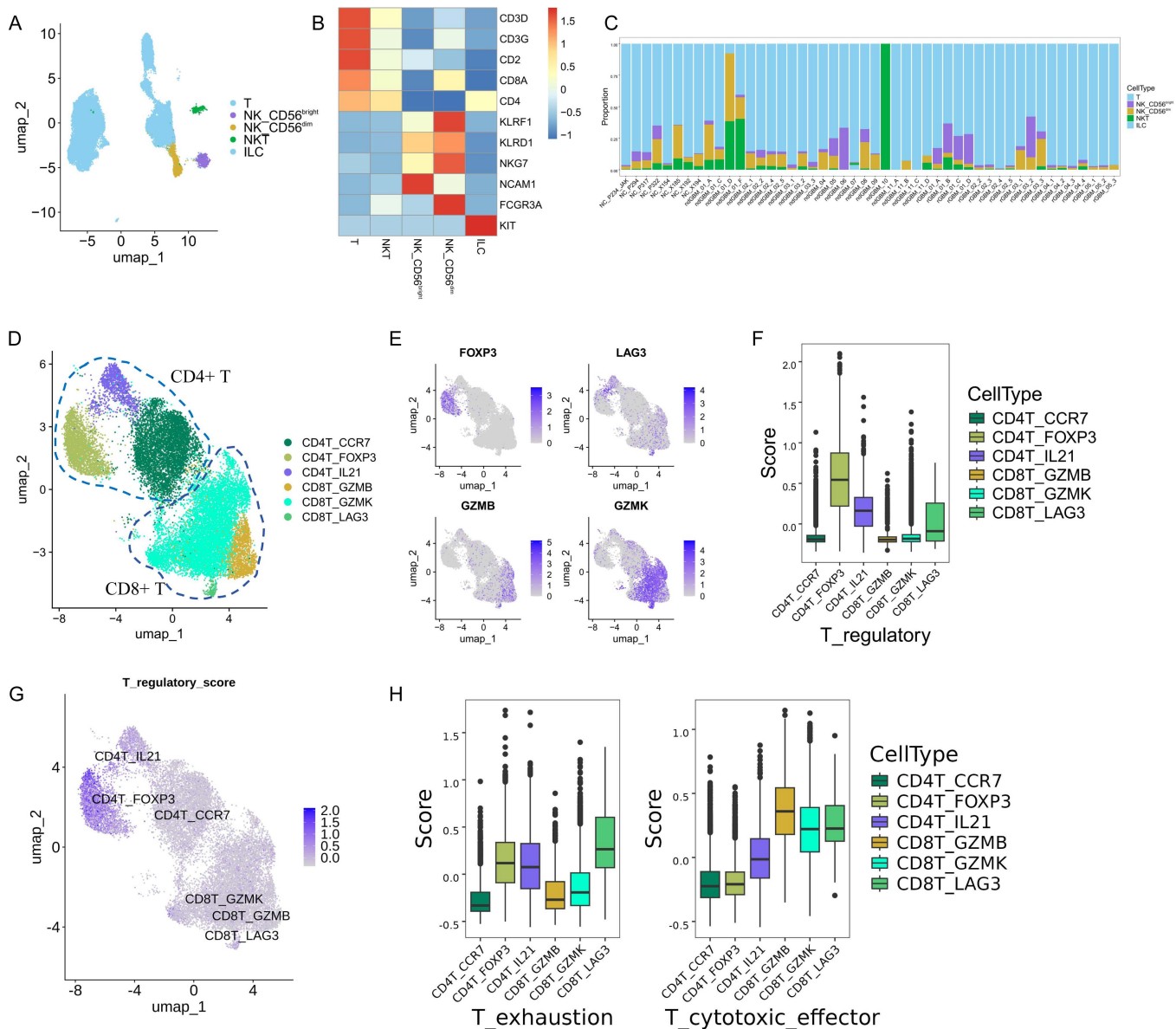

**Fig 3. Cellular composition of T and NK cells.** (A) UMAP plots of T and NK cells, with each cell color coded to indicate the associated cell types. (B) Heatmap displayed the classic markers used for T and NK cell type annotation. (C) T_NK cell type composition of each tumor fragment and dataset. (D) UMAP plots of CD4 + T and CD8 + T cells, with each cell color coded to indicate the associated cell types. (E) FeaturePlot displayed the classic markers used for CD4 + T and CD8 + T subtype annotation. (F) T regulatory scores of six T cell subtypes. (G) FeaturePlot displayed the T regulatory scores in six T cell subtypes. (H) T exhaustion and cytotoxicity scores of six T cell subtypes.

subsets. The CD4T_FOXP3 cells, characterized by elevated *FOXP3*, *CTLA4*, and *TIGHT* expression, were identified as regulatory T (Treg) cells. The CD4T_CCR7 subset, with high levels of *CCR7* and *SELL*, was classified as Naive T cells. The CD4T_IL21 subset, known for significant *IL21* expression, was recognized as helper T cells (Fig 3D, 3E). The CD8 + T cells were also divided into three groups. The CD8T_LAG3 subset, distinguished by high expression of *LAG3*, *PDCD1*, and *TIGHT*, was identified as exhausted T (Trex) cells. The CD8T_GZMB subset, with robust expression of *GZMB*, *GZMA*, and *PRF1*, was classified as effector T (Teff) cells. The CD8T_GZMK subset, showing high levels of *GZMK*, was identified as effector

memory (Tem) T cells (Fig 3D, 3E). We utilized the AddModuleScore function to calculate regulatory, exhaustion, and cytotoxic scores for these T subsets, respectively. As expected, the Treg (CD4T_FOXP3) cells displayed significantly higher regulatory scores (Fig 3F, 3G). The tumor-infiltrating CD8T_LAG3 cells exhibited a notably higher exhaustion score, whereas the effector CD8T_GZMB subset showed relatively higher cytotoxicity scores. Moreover, the CD8T_LAG3 (Tex) cells also demonstrated the second-highest levels of cytotoxic effector signatures, indicating retained effector capabilities (Fig 3H). Overall, the T cells displayed a dysfunctional state, emphasizing the complex immune responses within the GBM TME.

## Detailed analysis of NK cells uncovering CD56dim_DNAJB1 dysfunctional state

Next, we aimed to delineate the specific characteristics of NK cells within the TME of GBM. Two distinct NK cell types were identified based on the expression levels of *NCAM1* (CD56) and *FCGR3A* (CD16): CD56$^{bright}$CD16$^{lo}$ and CD56$^{dim}$CD16$^{hi}$ (Fig 3A). CD56$^{dim}$CD16$^{hi}$ NK cells primarily kill target cells through perforin and granzyme release, while CD56$^{bright}$CD16$^{lo}$ NK cells have immunoregulatory functions and produce cytokines. In our study, CD56$^{dim}$CD16$^{hi}$ NK cells showed increased expression of cytotoxic effector genes, while CD56$^{bright}$CD16$^{lo}$ NK cells exhibited upregulation of specific cytokines, as expected.

Further analysis subdivided the CD56$^{bright}$CD16$^{lo}$ NK cells into two subsets: NK_CD56$^{bri}$_KLRK1 and NK_CD56bri_CXCR6 (Fig 4A, 4B). The NK_CD56$^{bri}$_KLRK1 subtype is distinguished by elevated levels of *KLRK1*, also known as NKG2D. *KLRK1* is essential for enabling NK cells to activate, recognize, and eliminate tumor cells by interacting with stress-induced ligands on transformed cells [40]. This interaction is vital for triggering NK cells to attack and destroy tumor cells, a crucial mechanism for controlling tumor growth and promoting tumor regression [41,42]. The NK_CD56$^{bri}$_CXCR6 subtype is notable for expressing cytokine genes such as *CXCR6*, *CCL4*, *CCL4L2*, *CCL3*, and *CCL3L1*. High expression of *CXCR6* likely contributes to the increased infiltration of NK cells into the TME, facilitating their intense activation and potentially enhancing antitumor activity [42].

Similarly, the CD56$^{dim}$CD16$^{hi}$ NK cells were classified into NK_CD56$^{dim}$_CCL3 and NK_CD56$^{dim}$_DNAJB1 subtypes (Fig 4C, 4D). The NK_CD56$^{dim}$_CCL3 subtype, characterized by high levels of *CCL3*, likely enhances antitumor immunity by promoting the recruitment of NK cells and other immune cells, such as DC and CD8 + T cells, thereby facilitating increased Interferon-gamma(IFN-γ) production. This mechanism is crucial for activating and recruiting additional immune cells to the tumor site, potentially enhancing the efficacy of therapies such as anti-PD-1. The CD56$^{dim}$_DNAJB1 subset was particularly noted for expressing stress response genes such as *DNAJB1*, *HSPA1A*, and *HSP90AA1*. This subset also showed elevated levels of *NR4A1*, identified as a key mediator of T cell dysfunction, aligning with characteristics of the recently described TaNK population [43].

We further analyzed the gene expression signature to investigate the functional differences between CD56$^{dim}$_CCL3 and CD56d$^{im}$_DNAJB1 subtypes. Notably, the CD56dim_DNAJB1 subset exhibited relatively higher cytotoxicity and stress scores (Fig 4E). This suggests a unique role for the CD56$^{dim}$_DNAJB1 subset in stress response and inflammation, potentially influencing the overall functionality of the NK cell population. To investigate the specific enriched signaling pathways, we performed GSEA analysis using the KEGG and Hallmarks gene set from the MsigDB. Our analysis indicated that compared to CD56$^{dim}$_CCL3, CD56$^{dim}$_DNAJB1 showed significant upregulation in PD-L1 and PD-1 checkpoint pathway and HALLMARK_MTORC1_SIGNALING pathway (Fig 4F). Recent studies have shown that activation of the PD-1 receptor hinders the cytotoxic functions of NK cells, leading to

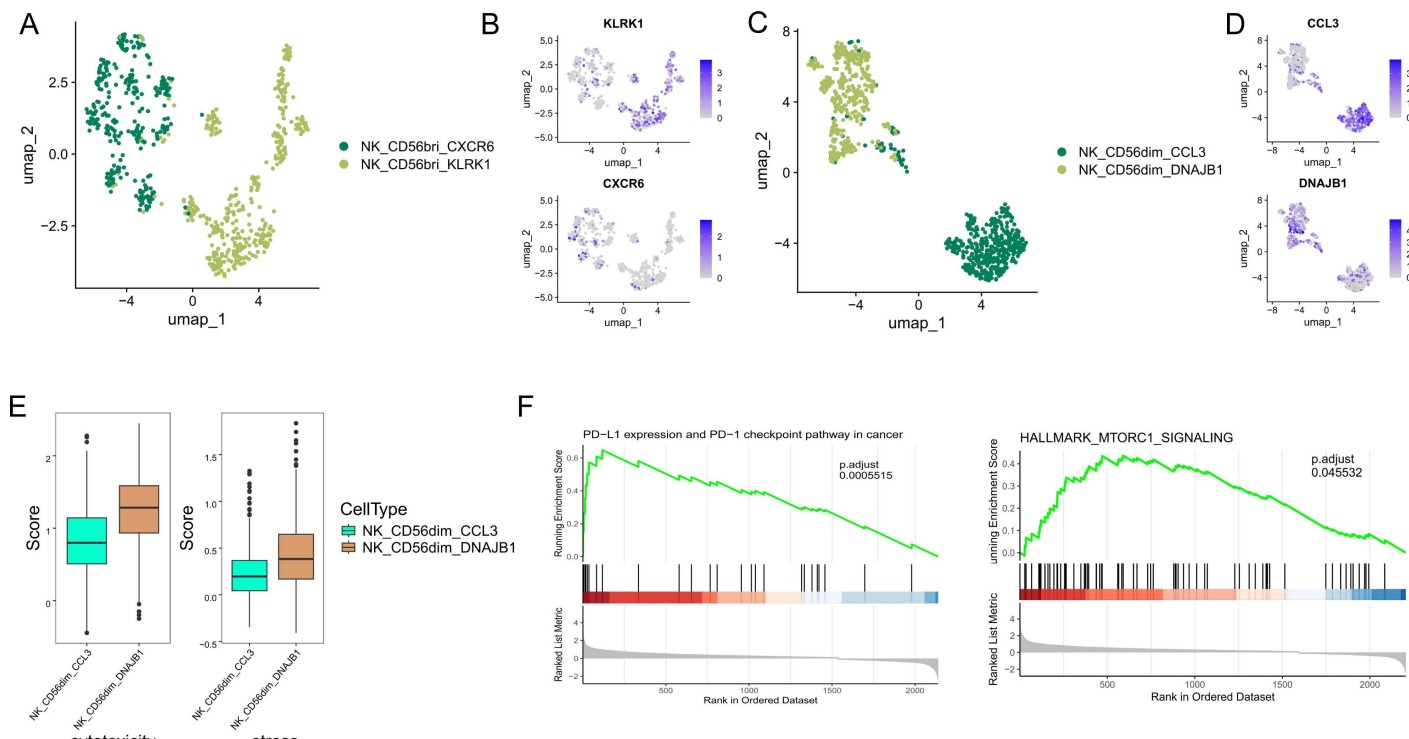

**Fig 4. Cellular composition of NK cells.** (A) UMAP plots of CD56bright NK cells, with each cell color coded to indicate the associated subtypes. (B) FeaturePlot displayed the classic markers used for CD56bright NK subtype annotation. (C) UMAP plots of CD56dim NK cells, with each cell color coded to indicate the associated subtypes. (D) FeaturePlot displayed the classic markers used for CD56dim NK subtype annotation. (E) NK cytotoxicity and stress scores of two CD56dim NK cell subtypes. (F) Different enrichment of KEGG and 'Hallmark' pathways between CD56dim_DNAJB1 and CD56dim_CCL3 by GSEA analysis.

immunosenescence. The PD-1/PD-L1 checkpoint pathway interacts with different immunosuppressive cells, including Tregs, myeloid-derived suppressor cells, and M2 macrophages. Furthermore, the mTOR signaling pathway is crucial in enhancing PD-L1 protein expression and facilitating immunosuppression [44].

## Complex intercellular communication networks in the GBM TME

To investigate cell-cell interactions and specific ligand-receptor pairs within the GBM TME, we performed CellphoneDB analyses, focusing on glioma alongside five TAM and four NK cell subtypes. Our analysis revealed significant communication between tumor and immune cells, particularly notable interactions between glioma cells and three TAM subsets: TAM_NLRP3, TAM_MRC1, and TAM_ISG15 (Fig 5A, 5B). Our results demonstrated that glioma cells exhibit elevated levels of amyloid precursor protein (APP) (Fig 5C). Additionally, high expression of CD74 was observed across five TAM subtypes, establishing a specific APP-CD74 ligand-receptor interaction. Research in various cancer models has highlighted that CD74+ TAMs can undermine immune surveillance, diminish the effectiveness of immune responses, and foster an immunosuppressive TME (Fig 5C). Furthermore, the analysis indicated high expression of Amphiregulin (AREG), a member of the epidermal growth factor family, in four NK cell subsets. Epidermal Growth Factor Receptor (EGFR), which serves as the receptor for AREG as well as for EREG and HBEGF, was prominently expressed in glioma cells (Fig 5D). Previous research have shown that interactions mediated by EGFR and AREG may augment the invasiveness and metastatic potential of tumors [45,46]. We also evaluated several

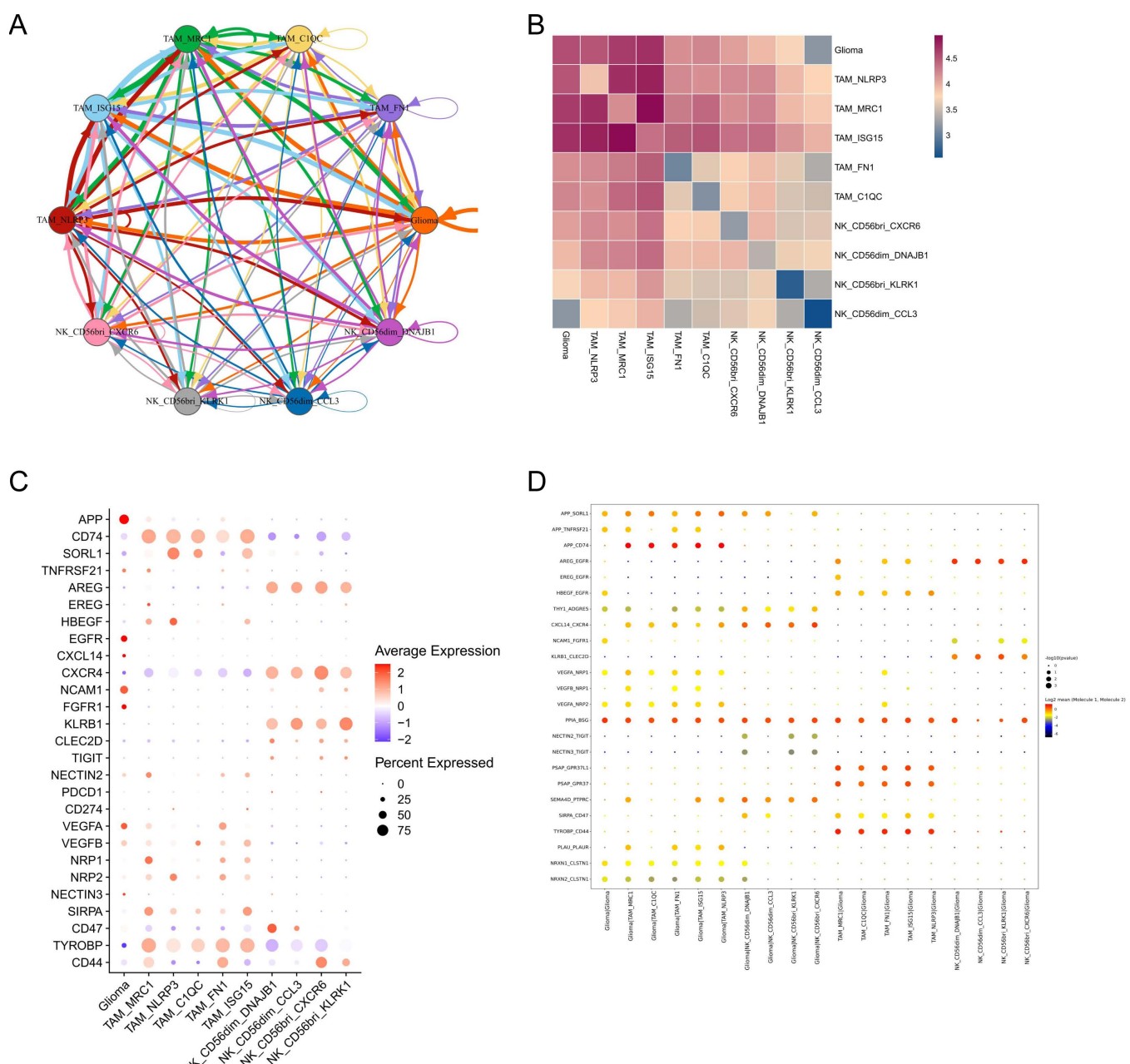

**Fig 5. Cell-cell communication among glioma and immunity cell types.** (A) Capacity for intercellular communication between malignant cells and immune cells. Each line color indicates the ligands expressed by the cell population represented in the same color (labeled). The lines connect to the cell types that express the cognate receptors. (B) Heatmap displayed the communication strength between two cell types. (C) Dot plot showing the average expression of highlighted ligands and receptors across cell types used for CellphoneDB analysis. (D) Cell-cell communication analysis using CellphoneDB. Depicted are the dot plots of ligand-receptor pairs for glioma-immunity and immunity-glioma signaling.

inhibitory receptors on the surface of NK cells, including *NKG2A*, *KLRB1* (CD161), *CLEC2D*, *TIGIT*, and *PDCD1*, suggesting the impaired cytotoxic activity of NK cells in the TME. This comprehensive analysis underscores the complex interplay of immune evasion mechanisms within the GBM TME, revealing potential targets for therapeutic intervention.

## Discussion

In this study, we have collected a large dataset of scRNA datasets of GBM, consisting of 48 samples from 24 GBM patients. The large dataset provides a strong foundation for our comprehensive analysis of tumor-infiltrating TAM and NK cells, which displayed functional diversity. Additionally, we uncovered the complex interactions between malignant glioma cells and TAM and NK cell subtypes in the TME. Through this comprehensive analysis, our study provides valuable insights into the role and dynamics of immunity cells within the GBM TME.

TAMs play a crucial role in regulating tumor progression and immune evasion. It is established that M1-like macrophages exhibit anti-tumor activities, whereas M2-like macrophages promote tumor growth. During the initial phase of GBM formation, monocyte-derived macrophages (MDMs) characterized by chemokine receptor 2 (CCR2) are recruited to perivascular regions, while TAMs enriched with CX3CR1 are recruited to peritumoral areas [47]. However, as the tumor microenvironment induces a shift towards a pro-invasive and immunosuppressive M2 phenotype, TAMs rapidly transition to an M2-like state, promoting tumor progression [48–50]. Consistent with previous findings, TAM_ISG15 macrophages exhibited relatively higher M1 signatures, suggesting their potential role in enhancing immune responses. Therefore, converting M2-like macrophages to M1-like macrophages is critical for the effectiveness of macrophage-based therapies.

Furthermore, the IFN-γ is instrumental in the immune system's response to various challenges, including tumor proliferation. This is critically involved in M1 macrophage polarization within tumor tissues. Yuan et al. (2022) [51] reported that RGS12 plays a significant role in the polarization and activation of M1 macrophages, driven by the interferon-gamma response, within the bone marrow microenvironment of multiple myeloma patients. Understanding the mechanisms behind M1 and M2 macrophage polarization and their contributions to tumor progression allows researchers to develop more targeted therapeutic strategies aimed at disrupting these pathways and thus potentially inhibiting cancer growth [51].

CAR-NK therapies have proven to be highly effective in treating blood cancers like lymphoma, myeloma, and leukemia [19,20]. However, their ability to combat solid tumors is hindered by the complex nature and unfavorable conditions within the TME. The dysfunction of CD8 + T cells within these environments is a known factor contributing to the limited success of CAR-T therapies, with various inhibitory receptors like PD-1, CTLA4, TIGIT, and TIM3 playing a role [43]. Yet, the impact of these immune checkpoints on NK cell dysfunction remains unclear. In our study, we observed potential dysfunction in tumor-infiltrating NK cells. These NK subtypes exhibited low expression of *B3GAT1* (CD57) and the killer immunoglobulin-like receptor (KIR) family, indicating immaturity and reduced killing capacity. This dysfunction of NK cells, similar to T cell exhaustion, suggested impaired natural cytotoxicity in GBM. In addition, we also identified a unique subset of NK cells, CD56dim_DNAJB1, which shares similarities with the recently discovered TaNK cells. TaNK cells have been associated with poor patient outcomes and resistance to immunotherapy in different cancer types [43]. The CD56dim_DNAJB1 subset in GBM displayed significant cytotoxic capabilities and appeared to be under considerable stress, indicating a potential involvement in the immune response against tumors. Interestingly, these cells exhibited low expression of immune checkpoint genes such as *PDCD1* and *CTLA4*, suggesting they may not respond to anti-PD-1/CTLA-4 therapies. Given their distinct characteristics compared to Tex cells and existing immune checkpoint blockade treatments, we propose that the CD56dim_DNAJB1 subset could impact or mirror immune responses in GBM.

The other three NK subsets may also play complex roles and potentially enhance the outcomes of immunotherapeutic strategies in GBM. Elevated levels of CCL3 in NK cells may significantly improve antitumor immunity by promoting the recruitment of NK cells and other immune cells to the tumor site. *KLRK1*, also known as NKG2D, is a critical activating receptor on NK cells. This receptor is pivotal in mediating immune responses, recognizing stress ligands frequently upregulated in cancer cells due to cellular stress or transformation [41]. The interaction between KLRK1 and these ligands prompts NK cells to attack and destroy tumor cells, a vital process in controlling tumor growth and promoting regression [41,42]. Additionally, the activation of KLRK1 is essential for the effective infiltration and activity of NK cells in tumors, as evidenced by studies showing enhanced NK cell infiltration and activation in tumor tissues following lenvatinib treatment [52]. *KLRK1* also recruits Ena/VASP-like (EVL) to the cytotoxic synapse in NK cells, facilitating F-actin generation and enhancing cytotoxicity through the Grb2-VAV1 signaling pathway [40]. Notably, *KLRK1* targets cancer stem cells (CSCs) within tumors, with NKG2D-activated NK cells preferentially attacking CSCs, which are often resistant to conventional therapies and play a crucial role in tumor recurrence and metastasis [53]. These discoveries underscore the significant role of *KLRK1* in NK cell-mediated tumor immunity and its potential as a therapeutic target.

High expression of *CXCR6* in NK cells within tumor tissues suggests multiple important roles in the TME, impacting immune cell interactions and potential therapeutic outcomes. Elevated levels of *CXCR6* may enhance the efficacy of immunotherapies such as IL-12 and PD-1 blockade by promoting T cell-NK cell-dendritic cell cross-talk, essential for robust antitumor immunity [22]. This expression could reflect unique phenotypic and functional traits, including increased cytokine production and enhanced degranulation. Conversely, it may also play a role in immunosuppression within the tumor, affecting other immune cells and potentially contributing to a tolerogenic microenvironment. For instance, high CXCR6 expression has been associated with poor prognosis in certain cancers like esophageal squamous cell carcinoma, influencing pathways such as IL-6-induced CD39 expression and could serve as a biomarker to predict patient outcomes and tailor treatments [54]. Thus, the complex role of *CXCR6* expression in NK cells highlights the intricate interplay between NK cell function, tumor immune evasion, and therapeutic responses, guiding future research into targeted therapies and biomarker development.

This study reveals the functional heterogeneity of glioma and immune cells within TME, providing potential targets for developing immunotherapeutic strategies against GBM. These findings have significant clinical implications, as they contribute to the design of more precise immunotherapy approaches, which could potentially improve patient outcomes and prognosis in the future. Certainly, certain inevitable limitations should be noted in this work. Firstly, our research data were derived from public databases, and some of the data parameters might be incomplete, leading to the potential risks of error and deviation. Secondly, this study is the lacks of comprehensive clinical information, such as patient age, genetic markers, and tumor location. This limitation has restricted our ability to perform subgroup analyses that could provide more nuanced insights into the differential effects of the interventions studied. Additionally, our findings need to be further validated through Larger sample sizes or even prospective studies are needed to confirm these findings. We are considering further research on this in the future.

In summary, our comprehensive analyses have advanced the current understanding of the TME in GBM. We identified various subtypes of tumor-filtrating TAMs and NK cells, highlighting their robust interactions with glioma cells. These insights could inform the development of more effective macrophage- and NK cell-based immunotherapeutic strategies for GBM.

## Supporting information

**S1 Fig. Dissection of the malignant glioma cells.** (A) UMAP plots of malignant glioma cells identify in this study, with each cell color coded to indicate the associated dataset. (B) UMAP plots of malignant glioma cells identify in this study, with each cell color coded to indicate the associated clusters.
(PNG)

**S2 Fig. Dissection of the M1 polarization and M2 polarization in TAM cells.** (A) Dot plot showing the average expression of highlighted M1-like and M2-like genes in all TAM clusters. (B) FeaturePlot displayed the M1 polarization scores in five TAM cell subtyps. (C) FeaturePlot displayed the M2 polarization scores in five TAM cell subtyps.
(PNG)

**S1 Table. Clinical information of the GBM patients from two published articles.**
(XLSX)

**S2 Table. The genesets of M1 polarization and M2 polarization.**
(XLSX)

## Author contributions

**Data curation:** Han Cheng, Yan Yan, Zhuolin Ma, Siwen Fu, Zhi Ji, Ziyi Zou.

**Formal analysis:** Zhuolin Ma, Zhi Ji.

**Methodology:** Han Cheng, Siwen Fu.

**Resources:** Yan Yan.

**Software:** Han Cheng, Yan Yan, Biao Zhang, Zhuolin Ma, Siwen Fu.

**Supervision:** Biao Zhang, Qin Wang.

**Visualization:** Qin Wang.

**Writing – original draft:** Han Cheng, Yan Yan, Siwen Fu, Zhi Ji, Ziyi Zou, Qin Wang.

**Writing – review & editing:** Ziyi Zou, Qin Wang.

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
