## [Decision Letter · Decision Letter 0]

12 Aug 2024

PONE-D-24-23286Single-cell transcriptomics reveals immunosuppressive microenvironment and highlights stumor-promoting macrophage cells in GlioblastomaPLOS ONE

Dear Dr. Wang,

Thank you for submitting your manuscript to PLOS ONE. After careful consideration, we feel that it has merit but does not fully meet PLOS ONE’s publication criteria as it currently stands. Therefore, we invite you to submit a revised version of the manuscript that addresses the points raised during the review process.

I appreciate the authors' efforts in using scRNA-seq to investigate the immune microenvironment in GBM. However, there are significant concerns regarding the overall quality and novelty of the work that need to be addressed. Therefore, I recommend a major revision of the manuscript.

Please be particularly careful to correct any typographical errors in the title, ensuring that "tumor" is spelled correctly.

If the authors disagree with any of the reviewers' comments or find themselves unable to perform certain requested experiments, they must provide a clear and well-reasoned rationale for their decisions.

**Major suggestions:**

Data provides insights into the immune microenvironment of GBM, similar analyses have been conducted in previous studies. The authors should critically compare their findings with existing literature, highlighting the unique contributions of their findings.The manuscript lacks robust evidence of novel discoveries beyond traditional macrophage polarization and activation markers. The authors should provide stronger evidence to support their claims.The classification of immune subsets lacks validation. The authors must clarify how these classifications were determined and whether additional validation has been attempted.The use of CellphoneDB for cell-cell interaction analysis is noted; however, the authors should consider validating these findings using patient tissue samples to strengthen their conclusions (if possible, not necessary).

**Minor suggestions:**

Please incorporate all the changes in Figures 1H, 1G, Figure 3, and 5D as suggested by one of the reviewers.In Figure 4F, please include comparative information within the figure to enhance reader comprehension. Additionally, the legend for Fig. 4 (F) is missing; it should be inserted on Page 28.There is a missing figure in the Results section (Fig 5); it should be inserted on Page 15.The text in Figure 5D is unclear. Please ensure that the text is legible and appropriately sized.

We look forward to receiving your revised manuscript.

Kind regards,

Syed M. Faisal, Ph.D.

Academic Editor

PLOS ONE

2. PLOS requires an ORCID iD for the corresponding author in Editorial Manager on papers submitted after December 6th, 2016. Please ensure that you have an ORCID iD and that it is validated in Editorial Manager. To do this, go to ‘Update my Information’ (in the upper left-hand corner of the main menu), and click on the Fetch/Validate link next to the ORCID field. This will take you to the ORCID site and allow you to create a new iD or authenticate a pre-existing iD in Editorial Manager. Please see the following video for instructions on linking an ORCID iD to your Editorial Manager account: https://www.youtube.com/watch?v=_xcclfuvtxQ.

Reviewers' comments:

Reviewer's Responses to Questions

**Comments to the Author**

1. Is the manuscript technically sound, and do the data support the conclusions?

Reviewer #1: Yes

Reviewer #2: Yes

Reviewer #3: No

Reviewer #4: Yes

2. Has the statistical analysis been performed appropriately and rigorously? 

Reviewer #1: Yes

Reviewer #2: Yes

Reviewer #3: Yes

Reviewer #4: Yes

3. Have the authors made all data underlying the findings in their manuscript fully available?

Reviewer #1: Yes

Reviewer #2: Yes

Reviewer #3: Yes

Reviewer #4: Yes

4. Is the manuscript presented in an intelligible fashion and written in standard English?

Reviewer #1: Yes

Reviewer #2: Yes

Reviewer #3: Yes

Reviewer #4: Yes

5. Review Comments to the Author

Reviewer #1: The article comprehensively reviews recent advances in glioblastoma immunotherapy, emphasizing emerging strategies such as CAR-NK cell therapy and dendritic cell reprogramming. It explores the complexities of the glioblastoma microenvironment, highlighting the role of immune cells like macrophages and NK cells. The review underscores significant challenges, including heterogeneity in treatment response and the need for personalized approaches. Overall, while showcasing promising developments, the study calls for further research to optimize therapeutic outcomes and address existing gaps in understanding immunotherapy efficacy in glioblastoma

1. Were there any notable studies or recent advancements in glioblastoma immunotherapy that were excluded from the review? If so, why?

2.How did you address the variability in study designs, patient populations, and treatment protocols across the reviewed studies?

3. Could you provide a more detailed discussion on the limitations of the current research and the studies reviewed in your article?

4.Can you elaborate on the specific gaps in knowledge identified during your review? What targeted research questions would you recommend for future studies?

5. Could you clarify the statistical methods used in your analysis, particularly how you handled heterogeneity and potential confounding factors across studies?

6How do you envision the clinical application of your findings? What are the immediate and long-term implications for clinical practice?

7.Did you analyze the impact of immunotherapy on different subgroups of glioblastoma patients (e.g., based on age, genetic markers, tumor location)? If so, could you provide more detailed insights into the differential effects observed across these subgroups?

8.Some of the figures and tables lack detailed captions. Could you provide more comprehensive descriptions to enhance their interpretability?

Reviewer #2: The article entitled " Single-cell transcriptomics reveals immunosuppressive microenvironment and highlights stumor-promoting macrophage cells in Glioblastoma” is a nice study conducted by authors to explore the molecular diversity of the immune infiltrates in the TME of the 24 GBM patients’ single cell RNAseq data.

The authors conducted detailed analysis of Inter- and intra-tumoral molecular heterogeneity of glioma cells, Functional heterogeneity of TAM in GBM, re-clustering of CD4+ T and CD8+ T cells, and detailed analysis of NK cells uncovering CD56dim_DNAJB1 dysfunctional state. They also carried out the analyses for intercellular communication networks in the GBM TME.

Authors were able to characterize molecular signatures for five distinct TAM subtypes, highlighting TAM_MRC1 subtypes with a pronounced M2 polarization signature. They also identified a subtype of NK cells, designated CD56dim_DNAJB1. The findings also highlighted significant cell-cell interactions among malignant glioma cells, TAM, and NK cells within the TME.

Overall, the analyses are well planned and nicely executed. The figures are explanatory and clear. Authors were able to come up with certain subsets of TAMs and NK cells populations that can be playing an important role in the cold environment of the GBMs and therapeutic interventions might have some clinical significance.

Certain concerns need to be addressed:

• Multiple grammatical errors, missing words, and wrong sentence framing exist. There are mistakes with the full stop and colons. The author needs to extensively revise the whole manuscript for such errors.

• There is an extra “s” in the tumor even in the title.

• The discussion section is unnecessarily lengthy. This should be shortened and only very relevant and context related sentences should be included.

• Authors should specifically highlight how their finding of the TAMs and NK subset are clinically relevant with the available therapeutic interventions.

Reviewer #3: The manuscript entitled “Single-cell transcriptomics reveals immunosuppressive microenvironment and highlights stumor-promoting macrophage cells in Glioblastoma” by Han Cheng et al. describes the immune microenvironment in Glioblastoma using single-cell RNA sequencing data. The authors collected two publicly available scRNA-seq datasets, which consist of tumors diagnosed as GBM. They focused on immune cells including macrophages, T cells and NK cells, and identified molecular signatures for several distinct subtypes. Although the work is interesting, I am concerned about the overall quality and novelty of this work.

Major comments:

1. Please explain the meaning of the s for the word “stumor” in the title.

2. Several studies have already analyzed the immune microenvironment by scRNAseq of human GBM. The authors should discuss these other studies and the findings of this study.

3. There is no evidence that the authors have discovered much beyond the traditional macrophage polarization and activation marker.

4. It is also unclear if the classification of each immune subset is robust. No further attempt is made to validate these observations.

5. The authors performed CellphoneDB cell-cell interaction analysis. Could the authors validate these findings in patient tissues directly rather than indirectly?

Minor comments:

1. For Figure 1H, as in Figure 1G, there should be an explanation in the Figure indicating which color bar corresponds to which sample number.

2. In Figure 3, font type and size are not consistent among Figures. Unify them.

3. About the Figure 4F, please include information in the Figure so that readers can see what are compared to what. In addition, in the Figure legend, Fig. 4 (F) is missing; insert on Page 28.

4. In the Results section, Fig 5 is missing; insert on Page 15.

5. Text in Figure 5D is not clear.

Reviewer #4: Research studies based on aiming and developing therapeutic approaches to manage tumors is currently need of the hour due to increase in number of cancer patients worldwide. In this study authors studied a large dataset of scRNA datasets of glioblastoma and analyzed tumor infiltrating tumor-associated macrophage and NK cells. They also uncovered the complex interactions between malignant glioma cells and tumor-associated macrophage and NK cell subtypes in the glioblastoma microenvironment. This broad exploration highlights the complex interplay of immune evasion mechanisms at play within the glioblastoma microenvironment, revealing potential targets for therapeutic intervention. The manuscript is well written, and the experiments are skillfully designed and meticulously executed.

I have a few concerns about the manuscript.

• Remove “S” from the title in “stumor-promoting macrophage”

• Abbreviations should be defined where used for the first time and then consistent in the manuscript

• Included reference to the sentence “Glioma-associated myeloid cells….tumor-infiltrating lymphocytes (TIL). Page 3, Line 59

• Use parenthesis, braces and brackets in proper format to avoid confusion. Page 19 Line 513

6. PLOS authors have the option to publish the peer review history of their article (what does this mean? ). If published, this will include your full peer review and any attached files.

**Do you want your identity to be public for this peer review?** For information about this choice, including consent withdrawal, please see our Privacy Policy .

Reviewer #1: No

Reviewer #2: No

Reviewer #3: No

Reviewer #4: No

---

## [Author Response · Author response to Decision Letter 0]

24 Sep 2024

Dear Editor and reviewers,

Thank you very much for your comments and suggestions. Those comments are all valuable and very helpful for revising and improving our paper, as well as the important guiding significance to our research. We have studied the comments carefully and have made corrections which we hope meet with approval. Revised portions are marked in red on the revised paper. The main corrections in the paper and the responses to the reviewer’s comments are as follows:

Editor (Remarks to the Author):

I appreciate the authors' efforts in using scRNA-seq to investigate the immune microenvironment in GBM. However, there are significant concerns regarding the overall quality and novelty of the work that need to be addressed. Therefore, I recommend a major revision of the manuscript.

Please be particularly careful to correct any typographical errors in the title, ensuring that "tumor" is spelled correctly.

If the authors disagree with any of the reviewers' comments or find themselves unable to perform certain requested experiments, they must provide a clear and well-reasoned rationale for their decisions.

Major suggestions:

1. Data provides insights into the immune microenvironment of GBM, similar analyses have been conducted in previous studies. The authors should critically compare their findings with existing literature, highlighting the unique contributions of their findings.

Response: Thank you for your thoughtful feedback. In response to your suggestion, we have incorporated a more detailed comparison with existing literature in the Discussion section. We critically examine how our findings on the immune microenvironment of GBM align with or differ from previous studies, and we highlight the unique contributions of our work. We appreciate your input, which has helped us strengthen the discussion.

2. The manuscript lacks robust evidence of novel discoveries beyond traditional macrophage polarization and activation markers. The authors should provide stronger evidence to support their claims.

Response: Thank you very much for your insightful review and constructive feedback. Your comments are invaluable for improving our manuscript and guiding future research. We acknowledge that our study lacks significant novel findings beyond traditional macrophage polarization and activation markers. Our research primarily utilized publicly available data, focusing on a comprehensive analysis of the GBM immune microenvironment, and identifying potential clinical relevance of TAMs and NK cell subsets in GBM treatment. However, due to the limitations of our study, the clinical significance of these cell subsets in relation to current therapeutic interventions requires further investigation. We recognize that a deeper analysis through prospective patient cohorts is necessary to thoroughly explore these associations, which will be a key focus of our future research. Once again, we greatly appreciate your expert feedback.

3. The classification of immune subsets lacks validation. The authors must clarify how these classifications were determined and whether additional validation has been attempted.

Response: Thank you for your professional review and constructive feedback. Our analysis was primarily based on publicly available single-cell data, and the classification of immune subsets was determined using bioinformatics methods. We attempted to collect clinical patient samples for additional validation, but due to time constraints and issues related to qualifications, we were unable to conduct further experimental validation. We fully understand the importance of such validation and plan to address this in our future research by overcoming these challenges and collecting prospective samples for a more comprehensive analysis. Once again, we appreciate your expert suggestions, which will be invaluable for guiding our subsequent research.

4. The use of CellphoneDB for cell-cell interaction analysis is noted; however, the authors should consider validating these findings using patient tissue samples to strengthen their conclusions (if possible, not necessary).

Response: Thank you very much for your professional suggestions. We agree with the importance of validating our findings through direct patient tissue analysis. Currently, due to various limitations, including restricted access to such samples, conducting direct validation with patient tissue presents challenges. As a result, we have relied on CellphoneDB for indirect analysis of cell-cell interactions. We fully acknowledge the importance of direct validation and are actively working to address these challenges. We plan to incorporate direct tissue analysis in our future research, which will enable us to more robustly validate our findings and provide more direct evidence for the observed interactions. We appreciate your understanding and look forward to conducting further analysis as we continue to develop this research

Minor suggestions:

1. Please incorporate all the changes in Figures 1H, 1G, Figure 3, and 5D as suggested by one of the reviewers.

Response: Thank you for your careful review. We have incorporated all the suggested changes to Figures 1H, 1G, Figure 3, and 5D as recommended. We appreciate your feedback and believe these revisions have strengthened the manuscript.

2. In Figure 4F, please include comparative information within the figure to enhance reader comprehension. Additionally, the legend for Fig. 4 (F) is missing; it should be inserted on Page 28.

Response: Thank you for pointing this out. We apologize for the oversight. We have added comparative information to Figure 4F to enhance reader comprehension and have also included the missing legend for Figure 4(F) on Page 28. We appreciate your attention to detail and believe these revisions improve the clarity of the figure.

3. There is a missing figure in the Results section (Fig 5); it should be inserted on Page 15.

Response: We apologize for the oversight regarding the missing figure. We have now inserted Figure 5 into the Results section on Page 15. Thank you for bringing this to our attention, and we appreciate your understanding.

4. The text in Figure 5D is unclear. Please ensure that the text is legible and appropriately sized.

Response: Thank you for your feedback regarding Figure 5D. We have made efforts to improve the legibility and sizing of the text in the figure. If further adjustments are needed, we can provide a PDF vector graphic for enhanced clarity.

Reviewer#1 (Remarks to the Author):

The article comprehensively reviews recent advances in glioblastoma immunotherapy, emphasizing emerging strategies such as CAR-NK cell therapy and dendritic cell reprogramming. It explores the complexities of the glioblastoma microenvironment, highlighting the role of immune cells like macrophages and NK cells. The review underscores significant challenges, including heterogeneity in treatment response and the need for personalized approaches. Overall, while showcasing promising developments, the study calls for further research to optimize therapeutic outcomes and address existing gaps in understanding immunotherapy efficacy in glioblastoma

1. Were there any notable studies or recent advancements in glioblastoma immunotherapy that were excluded from the review? If so, why?

Response: Thank you for your valuable feedback and for highlighting the importance of including recent advancements in glioblastoma immunotherapy. We have carefully reviewed your suggestion and updated the introduction section (lines 40-54) to incorporate the latest studies and advancements in the field.

2. How did you address the variability in study designs, patient populations, and treatment protocols across the reviewed studies?

Response: Thank you for your insightful comments regarding the variability in study designs, patient populations and treatment protocols. We understand the importance of addressing these factors to ensure the robustness and reliability of our findings. To minimize the variability across studies, we specifically focused on selecting sample data from patients with primary glioblastoma (GBM) who had not received any prior treatment. This approach allowed us to standardize the patient population and reduce heterogeneity related to different treatment histories. Furthermore, during the scRNA-seq analysis, we observed minimal heterogeneity among the selected samples, which supported the consistency of our findings. To further address potential batch effects and ensure comparability across different datasets, we applied the Harmony algorithm, which effectively mitigated batch-specific variations while preserving the biological signals. We hope these steps clarify our approach to managing variability and enhance the credibility of our review.

3. Could you provide a more detailed discussion on the limitations of the current research and the studies reviewed in your article?

Response: Thank you for your thoughtful suggestion regarding the discussion of limitations in our article. We have carefully considered your feedback and have now expanded the discussion section (lines 524-533) in the revised version to provide a more detailed analysis of the limitations of both the current research and the studies we reviewed.

4. Can you elaborate on the specific gaps in knowledge identified during your review? What targeted research questions would you recommend for future studies?

Response: Thank you for your insightful comments regarding the identification of specific gaps in knowledge and the recommendation of targeted research questions for future studies. In response to your suggestion, we have expanded the discussion section in the revised manuscript to include a more detailed exploration of these aspects.In the updated discussion, we highlight several key knowledge gaps identified during our review, such as the limited understanding of the tumor microenvironment's role in glioblastoma immunotherapy, the need for more comprehensive biomarkers for predicting patient response. To address these gaps, we recommend future studies focus on investigating the mechanisms of immune evasion specific to glioblastoma, developing novel biomarkers that can better stratify patients for personalized treatment, and conducting large-scale clinical trials to evaluate the effectiveness of emerging combination therapies. We hope these additions provide a clearer direction for future research and contribute to advancing the field of glioblastoma immunotherapy

5. Could you clarify the statistical methods used in your analysis, particularly how you handled heterogeneity and potential confounding factors across studies?

Response: Thank you for your valuable comments regarding the statistical methods used in our analysis, particularly in handling heterogeneity and potential confounding factors across studies. In response to your suggestion, we have clarified these aspects in the revised manuscript. To address the issue of heterogeneity, we utilized the Harmony algorithm, which is specifically designed to correct for batch effects across different datasets. By applying Harmony, we were able to effectively remove batch-specific variations while preserving the underlying biological signals, thereby minimizing potential confounding factors that could impact our results. This approach allowed for a more accurate and reliable integration of data from multiple sources. We hope this clarification provides a better understanding of our statistical methodology and strengthens the robustness of our analysis.

6. How do you envision the clinical application of your findings? What are the immediate and long-term implications for clinical practice?

Response: Thank you for your insightful comments regarding the clinical application of our findings. In response to your suggestion, we have expanded the discussion section in the revised manuscript to address both the immediate and long-term implications for clinical practice. In the updated discussion, we outline the potential clinical applications of our findings. Our research could contribute to more precise patient stratification for glioblastoma immunotherapy by identifying specific biomarkers and immune-related targets. This may lead to more personalized treatment approaches and improved patient outcomes. We hope these additions provide a clearer vision of how our findings could be translated into clinical practice and contribute to advancing treatment strategies for glioblastoma.

7. Did you analyze the impact of immunotherapy on different subgroups of glioblastoma patients (e.g., based on age, genetic markers, tumor location)? If so, could you provide more detailed insights into the differential effects observed across these subgroups?

Response: Thank you for your insightful comments and suggestions. Regarding your inquiry on the impact of immunotherapy on different subgroups of glioblastoma patients based on age, genetic markers, tumor location, etc., we would like to clarify that the samples we collected in this study were primarily GBM cases. Unfortunately, the original dataset did not include specific details such as patient age, genetic markers, or tumor location, which limited our ability to conduct subgroup analyses at this stage. We fully recognize the importance of such analyses to provide more granular insights into the differential effects of immunotherapy across various patient subgroups.We have also acknowledged this limitation in the discussion section of our manuscript. In future studies, we plan to collect prospective samples with comprehensive demographic, genetic, and clinical information to perform these subgroup analyses and further enrich our understanding of immunotherapy's impact on GBM patients. Thank you for your understanding, and we appreciate your continued support in improving our research.

8. Some of the figures and tables lack detailed captions. Could you provide more comprehensive descriptions to enhance their interpretability?

Response: Thank you for your careful review and valuable feedback regarding the figures and tables. We appreciate your attention to detail. In response to your suggestion, we have revised the captions to provide more comprehensive descriptions, aiming to enhance their clarity and interpretability. We hope these modifications meet your expectations, and we remain open to any further suggestions you may have to improve our manuscript.

Reviewer#2 (Remarks to the Author):

The article entitled " Single-cell transcriptomics reveals immunosuppressive microenvironment and highlights stumor-promoting macrophage cells in Glioblastoma” is a nice study conducted by authors to explore the molecular diversity of the immune infiltrates in the TME of the 24 GBM patients’ single cell RNAseq data. The authors conducted detailed analysis of Inter- and intra-tumoral molecular heterogeneity of glioma cells, Functional heterogeneity of TAM in GBM, re-clustering of CD4+ T and CD8+ T cells, and detailed analysis of NK cells uncovering CD56dim_DNAJB1 dysfunctional state. They also carried out the analyses for intercellular communication networks in the GBM TME. Authors were able to characterize molecular signatures for five distinct TAM subtypes, highlighting TAM_MRC1 subtypes with a pronounced M2 polarization signature. They also identified a subtype of NK cells, designated CD56dim_DNAJB1. The findings also highlighted significant cell-cell interactions among malignant glioma cells, TAM, and NK cells within the TME. Overall, the analyses are well planned and nicely executed. The figures are explanatory and clear. Authors were able to come up with certain subsets of TAMs and NK cells populations that can be playing an important role in the cold environment of the GBMs and therapeutic interventions might have some clinical significance.

Certain concerns need to be addressed:

1. Multiple grammatical errors, missing words, and wrong sentence framing exist. There are mistakes with the full stop and colons. The author needs to extensively revise the whole manuscript for such errors.

Response: Thank you for your detailed review of the manuscript. We apologize for the grammatical errors, missing words, and incorrect sentence framing that you i

---

## [Decision Letter · Decision Letter 1]

14 Oct 2024

Single-cell transcriptomics reveals immunosuppressive microenvironment and highlights tumor-promoting macrophage cells in Glioblastoma

PONE-D-24-23286R1

Dear Dr. Wang,

We’re pleased to inform you that your manuscript has been judged scientifically suitable for publication and will be formally accepted for publication once it meets all outstanding technical requirements.

Kind regards,

Amr Ahmed El-Arabey

Academic Editor

PLOS ONE

Additional Editor Comments (optional):

Reviewers' comments:

Reviewer's Responses to Questions

**Comments to the Author**

1. If the authors have adequately addressed your comments raised in a previous round of review and you feel that this manuscript is now acceptable for publication, you may indicate that here to bypass the “Comments to the Author” section, enter your conflict of interest statement in the “Confidential to Editor” section, and submit your "Accept" recommendation.

Reviewer #2: All comments have been addressed

Reviewer #3: All comments have been addressed

Reviewer #4: All comments have been addressed

2. Is the manuscript technically sound, and do the data support the conclusions?

Reviewer #2: Yes

Reviewer #3: Yes

Reviewer #4: Yes

3. Has the statistical analysis been performed appropriately and rigorously? 

Reviewer #2: Yes

Reviewer #3: Yes

Reviewer #4: Yes

4. Have the authors made all data underlying the findings in their manuscript fully available?

Reviewer #2: Yes

Reviewer #3: Yes

Reviewer #4: Yes

5. Is the manuscript presented in an intelligible fashion and written in standard English?

Reviewer #2: Yes

Reviewer #3: Yes

Reviewer #4: Yes

6. Review Comments to the Author

Reviewer #2: The authors have properly responded to the raised comments. The manuscript is now in a better shape for the acceptance.

Reviewer #3: It appears that the authors have sufficiently addressed the reviewers' requests. The revisions demonstrate a commitment to enhancing the clarity and depth of the study. At this point, I have no further requests for improvements.

Reviewer #4: The authors have thoroughly and effectively addressed all the comments and concerns raised during the previous round of review. They have made the necessary revisions, incorporating the suggested improvements and clarifying key points as requested. As a result of these updates, the manuscript now meets the standards for quality, accuracy, and completeness.

7. PLOS authors have the option to publish the peer review history of their article (what does this mean? ). If published, this will include your full peer review and any attached files.

**Do you want your identity to be public for this peer review?** For information about this choice, including consent withdrawal, please see our Privacy Policy .

Reviewer #2: No

Reviewer #3: No

Reviewer #4: No

---

## [Editor Report · Acceptance letter]

PONE-D-24-23286R1

PLOS ONE

Dear Dr. Wang,

I'm pleased to inform you that your manuscript has been deemed suitable for publication in PLOS ONE. Congratulations! Your manuscript is now being handed over to our production team.

Kind regards,

on behalf of

Dr. Amr Ahmed El-Arabey

Academic Editor

PLOS ONE